# Robust and efficient hydrogenation of carbonyl compounds catalysed by mixed donor Mn(I) pincer complexes

Wenjun Yang [1], Ivan Yu. Chernyshov[2], Robin K. A. van Schendel[1], Manuela Weber[3], Christian Müller[3], Georgy A. Filonenko [1✉] & Evgeny A. Pidko [1✉]

Any catalyst should be efficient and stable to be implemented in practice. This requirement is particularly valid for manganese hydrogenation catalysts. While representing a more sustainable alternative to conventional noble metal-based systems, manganese hydrogenation catalysts are prone to degrade under catalytic conditions once operation temperatures are high. Herein, we report a highly efficient Mn(I)-CNP pre-catalyst which gives rise to the excellent productivity (TOF° up to 41 000 h$^{-1}$) and stability (TON up to 200 000) in hydrogenation catalysis. This system enables near-quantitative hydrogenation of ketones, imines, aldehydes and formate esters at the catalyst loadings as low as 5–200 p.p.m. Our analysis points to the crucial role of the catalyst activation step for the catalytic performance and stability of the system. While conventional activation employing alkoxide bases can ultimately provide catalytically competent species under hydrogen atmosphere, activation of Mn(I) pre-catalyst with hydride donor promoters, e.g. KHBEt$_3$, dramatically improves catalytic performance of the system and eliminates induction times associated with slow catalyst activation.

[1] Inorganic Systems Engineering group, Department of Chemical Engineering, Faculty of Applied Sciences, Delft University of Technology, Van der Maasweg 9, 2629 HZ Delft, The Netherlands. [2] TheoMAT Group, ChemBio cluster, ITMO University, Lomonosova 9, St, Petersburg 191002, Russia. [3] Institute of Chemistry and Biochemistry, Freie Universität Berlin, Fabeckstraße 34/36, D-14195 Berlin, Germany. ✉email: G.A.Filonenko@tudelft.nl; E.A.Pidko@tudelft.nl

Catalytic hydrogenation of carbonyl derivatives with molecular hydrogen is an essential technique for the production of bulk and fine chemicals[1]. The state of the art in hydrogenation catalysis to this date is laid down by well-defined noble metal complexes based on ruthenium, iridium, and rhodium[2,3]. However, the requirements for more sustainable hydrogenation processes recently initiated a search for earth-abundant, inexpensive 3d metals that can replace their noble counterparts[4–7]. In this search, the catalysts based on highly biocompatible and abundant Mn metal became particularly prominent[8–13].

Manganese-based hydrogenation catalysis has become a subject of intense research since 2016, largely set off by the pioneering work of Beller and coworkers[14]. An Mn pincer complex **A** (Fig. 1) promoted hydrogenation of ketones, aldehydes and nitriles operating at 1–3 mol% loading at 60–120 °C and 10–50 bar $H_2$ pressure. Following the initial reports, the field of hydrogenation with Mn was extended to several prominent ligand platforms[9,10,15–27].

Specifically, in addition to aminopincer ligands, the diamino triazine-based pincers **B** and lutidine-derived PNN pincer **C** ligands were introduced to Mn-catalyzed hydrogenations by the groups of Kempe[28] and Milstein[29], and saw further improvement in recent years[30]. In addition to pincer ligands, several bidentate ligands have been employed in Mn catalysis. These include PN aminophosphines developed by our group (**D**, Fig. 1)[31], diphosphines **E** reported by the groups of Kirchner[32] and García[33]. The most recent addition to this set was reported by Sortais and coworkers who described catalyst **F** (Fig. 1) based on a bidentate ligand containing phosphine and N-heterocyclic carbene (NHC) donors. Together with catalyst **B**, complex **F** is one of the most potent Mn ketone hydrogenation catalysts requiring ca. 0.1 mol% catalyst loading for operation[34].

The activity of Mn catalysts is generally lower than that of noble metal catalysts with majority of Mn-catalyzed hydrogenations requiring relatively high catalyst loadings of 0.1–5 mol%—a feature that strongly limits their practical utility. We recently demonstrated that reliance on such high metal loadings in Mn catalysis might stem from the limited stability of Mn pre-catalysts, most noticeable when Mn loadings are low[35]. Namely, we noted that Mn(I)–NHC complexes featuring aminocarbene "CN" bidentate ligands were highly competent hydride transfer catalysts at low reaction temperatures and high metal loadings, but a rapid catalyst degradation took place upon even marginal increase of reaction temperatures or reduction of catalyst loading <100 p.p.m. with respect to reduction substrate[35]. Addressing the catalyst stability in this work, we developed an active and highly stable Mn(I) catalyst that can promote hydrogenation reactions at catalyst loadings as low as 5 parts per million. Responsible for such performance is the tridentate CNP ligand platform (Fig. 1) that exhibits highly unusual phosphine hemilability, and enables catalyst activation pathways unavailable for known Mn catalysts.

## Results

**Synthesis and hydrogenation activity of 3**. Our initial synthetic effort was targeted at addressing the stability of manganese catalysts utilizing bidentate "CN" ligands (**1**, Fig. 2) by extending the ligand with additional phosphine donor arm. This extension of CN ligand **1** was done via a straightforward reductive amination producing the air-stable **L1** in 81% yield (Fig. 2). The **L1** can undergo complexation to form **3** by a one-pot reaction involving pre-coordination to Mn(CO)$_5$Br followed by the base-assisted formation of the NHC complex (Fig. 2). Analytically pure MnCNP complex **3** was isolated in 51% yield with its identity confirmed by NMR and IR spectroscopy, and elemental analysis (see Supplementary Information).

The IR spectrum of **3** features three strong bands at 2021, 1943, and 1919 cm$^{-1}$ consistent with the presence of three carbonyl ligands in a *facial* arrangement within the cationic complex[15]. The characteristic $^{31}P$ resonances in NMR spectrum of **3** appear at $\delta = 37.5$ (s), $-144.4$ (hep, $^1J_{FP}$ 712.8 Hz) p.p.m. confirming the coordination of the phosphine donor arm and the presence of the hexafluorophosphate anion in **3**. Finally, the $^{13}C$ NMR revealed resonances at 217.7, 215.5, 213.9, and 187.0 p.p.m. confirming the presence of three inequivalent carbonyl ligands and an Mn-bound NHC ligand.

Complex **3** is a potent and stable precatalyst for ketone hydrogenation. We screened its performance in hydrogenation of acetophenone benchmark substrate in various solvents at different $H_2$ pressures (see Supplementary Tables 1 and 2), and found the $H_2$ pressure of 50 bar and dioxane solvent to be optimal for performance. This combination was further used to evaluate the impact of reaction temperature and catalyst loadings on the hydrogenation yield. The results listed in Table 1 indicate that the quantitative hydrogenation of acetophenone to the corresponding alcohol can be obtained with catalyst loadings as low as 50 p.p.m. at 60 °C (Table 1, entries 1–3). Importantly, catalyst **3** tolerates elevated reaction temperatures of 80 and 100 °C, that marks a significant improvement of thermal stability over the parent CN bidentate that rapidly degraded as the temperatures were elevated over 70 °C (ref. [35]). Even at 100 °C hydrogenations with **3** led to quantitative yields requiring only 50 p.p.m. catalyst loading (entries 5 and 6). At low catalyst loading conditions, the activity of **3** compares favorably with a related MnPNP system **A** (ref. [14];

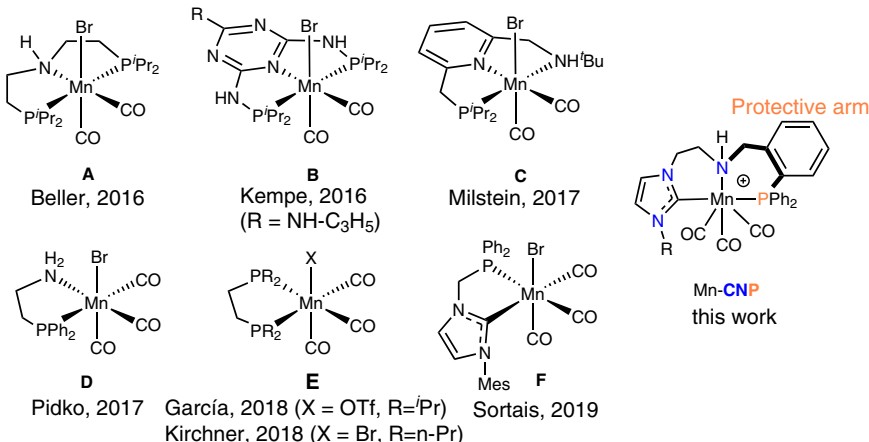

**Fig. 1 Mn complexes.** Selected examples of Mn hydrogenation catalysts and the complex used in this work.

**Fig. 2 Synthesis of Mn(I) complex 3. a** Conditions: **1** (8 mmol), **2** (1 eq.), NaBH(OAc)$_3$ (2.5 eq.), DCE (20 mL), rt, 12 h. **b** Conditions: (i) **L1** (0.5 mmol), MnBr(CO)$_5$ (1 eq.), THF (5 mL), 50 °C rt, 12 h; (ii) KHMDS (1.1 eq.), rt, 3 h.

**Table 1 Manganese-catalyzed hydrogenation of acetophenone[a].**

| Entry | Cat. (mol%/p.p.m.) | T (°C) | t (h)[b] | Yield (%)[c] |
|---|---|---|---|---|
| 1 | **3** (0.02/200) | 60 | 16 | 99 |
| 2 | **3** (0.01/100) | 60 | 16 | 99 |
| 3 | **3** (0.005/50) | 60 | 46 | 99 |
| 4 | **A** (0.02/200) | 80 | 24 | 67 |
| 5 | **3** (0.02/200) | 80 | 3 | 99 |
| 6 | **3** (0.02/200) | 100 | 1 | 99 |
| 7 | **3** (0.01/100) | 100 | 3 | 99 |
| 8 | **3** (0.005/50) | 100 | 28 | 99 |
| 9 | **3** (0.0025/25) | 100 | 28 | 87 |

[a]Conditions: acetophenone (5 mmol), Mn catalyst **3**, KO$^t$Bu (1 mol%), 1,4-dioxane (3 mL), $P$ = 50 bar H$_2$.
[b]Total reaction time and that of GC analysis, for H$_2$ uptake traces see Supplementary Information.
[c]Yield determined by GC with dodecane as internal standard.

entries 4 and 5). The hydrogenation with **3** at 200 p.p.m. loading at 80 °C is complete within 3 h, whereas catalyst **A** (Fig. 1) provides 67% conversion in 24 h under identical conditions.

The analysis of reaction progress using the H$_2$ uptake measurements (see Supplementary Information) reveals that catalyst deactivation at elevated temperatures is only pronounced at very low catalyst loadings. Namely, for the reaction at 100 °C (entries 6–9, Table 1), full ketone conversion can be reached with 50 p.p.m. of **3**, whereas at 25 p.p.m. the reaction does not proceed beyond 87% conversion level regardless of the reaction times employed (see Supplementary Fig. 38). Having observed that the introduction of a protective phosphine arm in complex **3** has markedly increased the catalyst thermal stability, we sought to improve the performance of **3** further. Apart from stability per se, we aimed at improving catalyst activation protocol that is an integral parameter to any catalytic system.

**Mechanistic analysis.** Catalyst activation, at large, is the reactivity pattern resulting in the generation of the active catalyst species. Similar to most bifunctional hydrogenation catalysts[4–13], our initial approach to catalyst activation involved the reaction of **3** with excess strong KO$^t$Bu alkoxide base followed by H$_2$ to form the catalytically active Mn–H moiety (Fig. 3a). Tracking this transformation with the IR spectroscopy we observed a rapid and clean conversion of **3** upon reaction with KO$^t$Bu (Fig. 3b) into the amido complex **4**. Notably, all three CO ligands were retained within **4** as follows from the presence of new bands at 1989, 1901, and 1885 cm$^{-1}$. The resonance of phosphine donor in **4** was slightly shifted upfield to $\delta$ = 33.6 p.p.m. in $^{31}$P NMR compared

to the initial cationic complex **3** ($\delta$ = 37.7 p.p.m.). At the same time complex **4**, as well as its parent complex **3** exhibited restricted mesityl group rotation dynamics evidenced by the loss of equivalency between *ortho*-methyl substituents of the mesityl group on the NMR timescales—a typical feature of Mn(I)–NHC complexes[35].

Complex **4** was stable in THF for up to 24 h and could be isolated as microcrystalline solid in 70% yield. The analysis of solid-state crystal structure of **4** confirmed the *facial* configuration of the tridentate CNP ligand implied by the NMR and IR spectral data (Fig. 3c). Remarkably, the single-crystal X-ray diffraction results revealed a highly unusual P-donor binding geometry in **4** with N–Mn–P angle of mere 67.4°. For comparison, the corresponding valent angles in related MnPNP pincer complexes are >80° (refs. [28–30]), indicating a significant coordination strain in complex **4** featuring the *fac*-bound CNP ligand.

Complex **4** reacts with H$_2$ gas upon heating, resulting in a loss of one of the CO ligands and the formation of isomers of a manganese hydride pincer complex **5** (Fig. 3). However, this reaction is particularly slow and proceeds to ca. 24% conversion of the starting complex **4** at 50 °C over 12 h under 3 bar H$_2$ pressure. The reaction gives rise to two new doublet resonances in $^1$H NMR spectrum at −3.46 and −3.49 p.p.m. with $^2J_{PH}$ = 60.0 and 68.0 Hz, respectively (Fig. 3d), corresponding to two isomers of dicarbonyl Mn–H species **5** with phosphine arm bound to Mn center. DFT analysis suggests that the main feature distinguishing these isomers is the respective positions of the axial carbonyl and hydride ligands relative to the meridionally bound CNP pincer

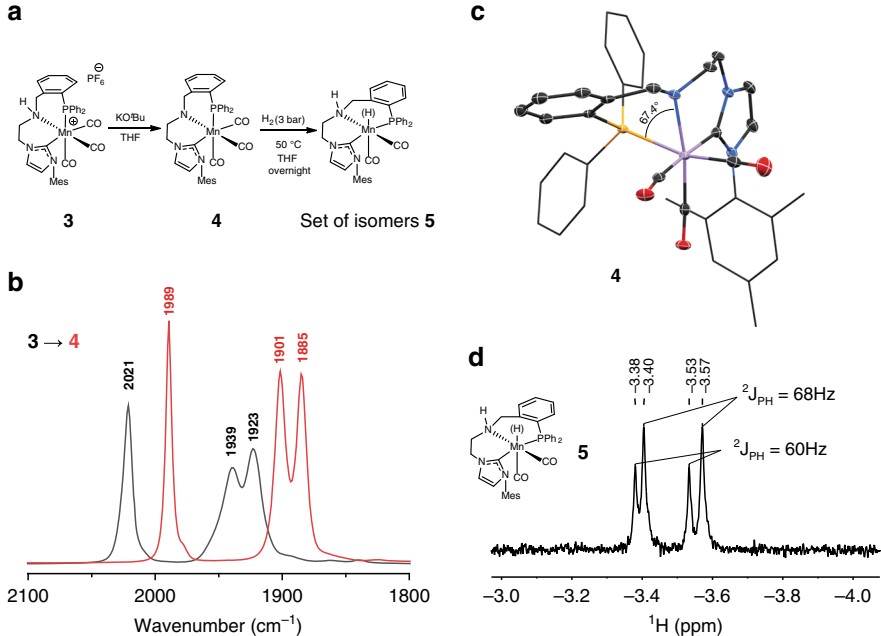

**Fig. 3 Activation and reactivity of complex 3. a** Generation of complex **5** upon reaction with $H_2$ (3 bar). **b** IR spectra of complex **3** (black) and in situ generated complex **4** (red) recorded in THF at 25 °C. **c** Molecular structure of complex **4** in the crystal with thermal ellipsoids drawn at 50% probability. **d** Hydride region of $^1H$ NMR (THF-$d_8$) spectra of in situ generated isomers of complex **5** (see Supplementary Fig. 19 for full spectra and analysis of isomers of **5**, conversion of **4**–**5** estimated at 24% by $^1H$ NMR).

(see Supplementary Figs. 40 and 41 for all analyzed structures), and allows ruling out the generation of *fac* isomers of the same composition.

From catalysis standpoint, the formation of Mn hydride species is generally accepted as a prerequisite for entering the hydrogenation cycle[8–13,16–19,29,30,33]. Our data, on the other hand, indicated that the Mn–H formation from **4** is slow and requires the irreversible loss of one CO ligand. Seeking for an alternative to the sluggish direct $H_2$ activation, we found that the reaction of precatalyst **3** with 2.5 equiv. of the KBHEt$_3$ hydride donor can also generate a Mn–H species in an instant manner. In THF-$d_8$ at room temperature the reaction of **3** with KBHEt$_3$ readily yields a reaction mixture containing 69% of the amido complex **4** with remainder comprised of new manganese hydride species **6** (Fig. 4a) that exist as a mixture of isomers. Unlike Mn hydride complex **5** observed in alkoxide-based activation protocol, species **6** features a free phosphine arm. The latter is evidenced by the appearance of the singlet resonances at $\delta = -16.2$ and $-16.6$ p.p.m. in $^{31}P$ NMR. As in the case of **5**, complex **6** exists as two isomers distinguished by $^{31}P$ resonances and those of hydride ligands appearing as singlets at $\delta = -3.92$ and $-4.40$ p.p.m. in $^1H$ NMR spectrum. Similar to the case of **5**, DFT analysis suggests that complex **6** exists with meridionally bound CNP ligand and hydride ligands occupying axial position within the complex (see Supplementary Fig. 41). Species **6**, being stable in solution for several hours, slowly convert to **5** as confirmed by in situ solution IR studies and NMR data depicted in Fig. 4a (see Supplementary Figs. 22–25 and 28 for full spectra).

An unusual feature of the CNP ligand, responsible for the formation of complex **6** is the apparent hemilabile nature of phosphine donor arm in MnCNP precatalyst. The phenomenon of ligand hemilability is often employed to rationalize reactivity of organometallic compounds[36–41], especially in the context of hybrid and multidentate ligands[42–44]. Invoked mainly for labile donor groups, e.g., oxygen or nitrogen[43], hemilability is scarce for phosphine donors in general[45–47] and for manganese phosphines in particular[48,49]. In case on MnCNP complexes, generation of

hydride complex **6** presents an attractive activation protocol for **3**. Unlike the sluggish base-assisted activation with molecular $H_2$, the reaction with KBHEt$_3$ proceeds instantly at room temperature and does not require CO ligand dissociation steps. The hydride species produces in such manner are catalytically competent and readily react with ketone substrates. Our stoichiometric studies indicate the higher reactivity of **6** toward ketones, compared to **5**. The stepwise introduction of a stoichiometric amount of acetophenone to the mixture containing **5** and **6** leads to a rapid disappearance of the resonances of **6**, highlighting it as a more competent hydride donor (see Supplementary Fig. 29).

We found that the improved activation protocol has a profound effect on the hydrogenation kinetics. Monitoring the hydrogenation kinetics, we could confirm that the use of KHBEt$_3$ promoter at 60 °C and 50 bar $H_2$ pressure significantly reduces the hydrogenation onset time compared to the KO$^t$Bu-promoted catalysis (Fig. 4b). While the activation of the MnCNP precatalyst with the alkoxide base resulted in ca. 15 min induction period, the KHBEt$_3$ treatment eliminated this lag time. Furthermore, the more selective precatalyst activation with the borohydride promotor resulted in a nearly threefold increase of the hydrogenation rate. We suggest this improvement to stem from an efficient catalyst activation protocol that allows for facile generation of competent hydride species **5** and **6**, thus ensuring the ability of Mn precatalyst to enter the catalytic cycle immediately. As our data depicted in Fig. 4a suggested that both **5** and **6** are exhibiting the hydride transfer reactivity upon the contact with acetophenone, we further attempted to observe the outcome of the catalytic turnover on the relative composition of the reaction mixture using NMR spectroscopy. Results of this experiment are presented in Fig. 4a. We observed that hydrogenation of acetophenone substrate with reaction mixtures containing predominantly hydride species **6** results in the accumulation of the dicarbonyl complex **5**. While suggesting that the catalytic turnover involving solely species **6** and associated ligand hemilability is possible, gradual accumulation of **5** in the course of several catalytic turnovers suggests that the

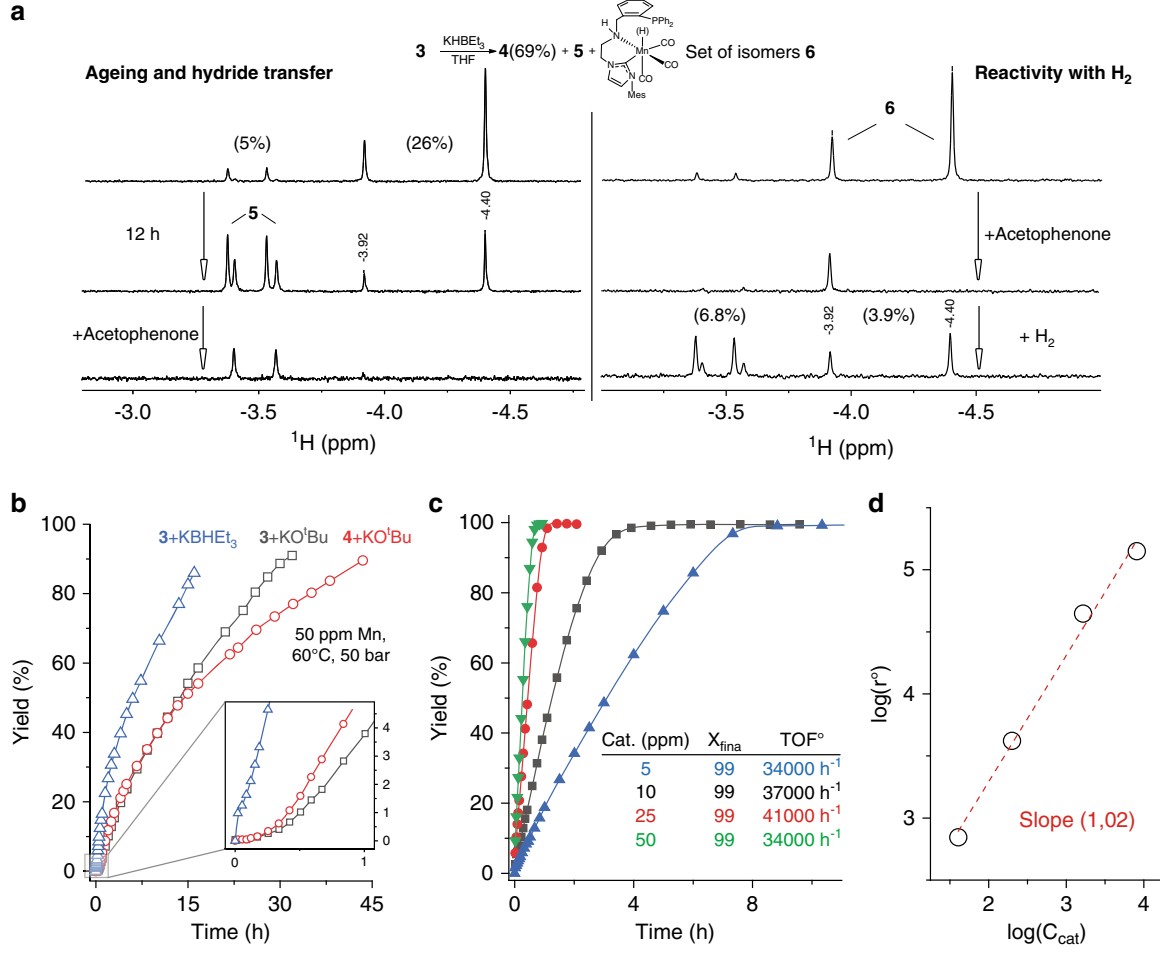

**Fig. 4 Catalyst activation with KBHEt₃ and corresponding catalytic performance. a** ¹H NMR spectra of the hydride region for KBHEt₃ activated complex **3**. Left panel shows slow conversion of **6** to complex **5** and subsequent reactivity with acetophenone; right panel shows reactivity of species **6** with acetophenone and regeneration of reactive hydride species in H₂ atmosphere (see Supplementary Figs. 32–35 for details); relative content of hydride species indicated in percent units of total Mn. **b** Kinetic traces for the hydrogenation of acetophenone with complexes **3** and **4** in the presence of 1 mol% KOtBu or 1 mol% KBHEt₃ promoters at 60 °C, 50 bar H₂, and 50 p.p.m. Mn loading. **c** Kinetic traces of acetophenone hydrogenation with **3** at different catalyst loading. Conditions: 50 bar H₂, 120 °C, 12.5 mmol substrate, 1 mol% of KBHEt₃, catalyst loading indicated on the graph. **d** Double logarithmic plot for reaction order analysis with respect to catalyst concentration for the data plotted in **c**.

**Table 2 Manganese-catalyzed hydrogenation of acetophenone with KBHEt₃ promotor[a].**

| Entry | Cat. 3 (mol%/p.p.m.) | $T$ (°C) | $t$ (h)[b] | Yield (%)[c] |
|---|---|---|---|---|
| 1 | 0.005/50 | 100 | 4 | 99 |
| 2 | 0.005/50 | 120 | 3 | 99 |
| 3 | 0.0025/25 | 120 | 3 | 99 |
| 4 | 0.001/10 | 120 | 6 | 99 |
| 5 | 0.0005/5 | 120 | 9 | 99 |
| 6 | No Mn | 120 | 12 | Trace |
| 7[d] | 0.005/50 | 120 | 3 | 99 |

[a]Reactions were conducted with acetophenone (5 mmol), Mn catalyst **3**, KBHEt₃ (1 mol%) in 1,4-dioxane (3 mL), $P = 50$ bar H₂.
[b]Total reaction time and that of GC analysis, for H₂ uptake traces see Supplementary Information.
[c]Yield determined by GC with dodecane as internal standard.
[d]Reactions was conducted under conditions identical to entry 2 in presence of 2 mol% Hg.

reaction temperatures (Table 2). The borohydride activation allowed for a sevenfold reduction of reaction time compared to the best example of alkoxide-promoted hydrogenation described above (Table 2, entry 1 vs. Table 1, entry 8). Furthermore, we could use **3** at 120 °C with the catalyst loading reduced from 50 to 5 p.p.m. without the loss of catalytic performance (Table 2, entries 2–5). Even at 5 p.p.m. loading acetophenone hydrogenation was brought to completion within 9 h at 120 °C. The homogeneous nature of Mn catalysis in this reaction was confirmed by control experiments (Table 2, entries 6 and 7). Our kinetic data collected for hydrogenations at 12.5 mmol scale indicated the first order in precatalyst **3** (Fig. 4d) with exceptional TOF° values of >40,000 h⁻¹ (Fig. 4c) under these reaction conditions.

**Substrate scope.** Finally, complex **3** proved to be a versatile hydrogenation catalyst (Fig. 5). With mere 50 p.p.m. Mn loading at 120 °C, aromatic ketones **8a–8i** were reduced in high to quantitative yield with the exception of sterically demanding *tert*-butyl phenyl ketone **8b** that was converted with 81% yield. Milder conditions (80 °C) were used for activated ketones with hetero-cycles and functional groups (**8j–8l**), affording corresponding alcohols with 85–99% isolated yields. Cyclic and linear aliphatic ketones **8m–8q** were also hydrogenated with quantitative yields.

hydrogenation can likely proceed over the complex **5** at low catalyst loadings.

The use of the improved activation method allowed carrying out the hydrogenations at lower catalyst loadings and higher

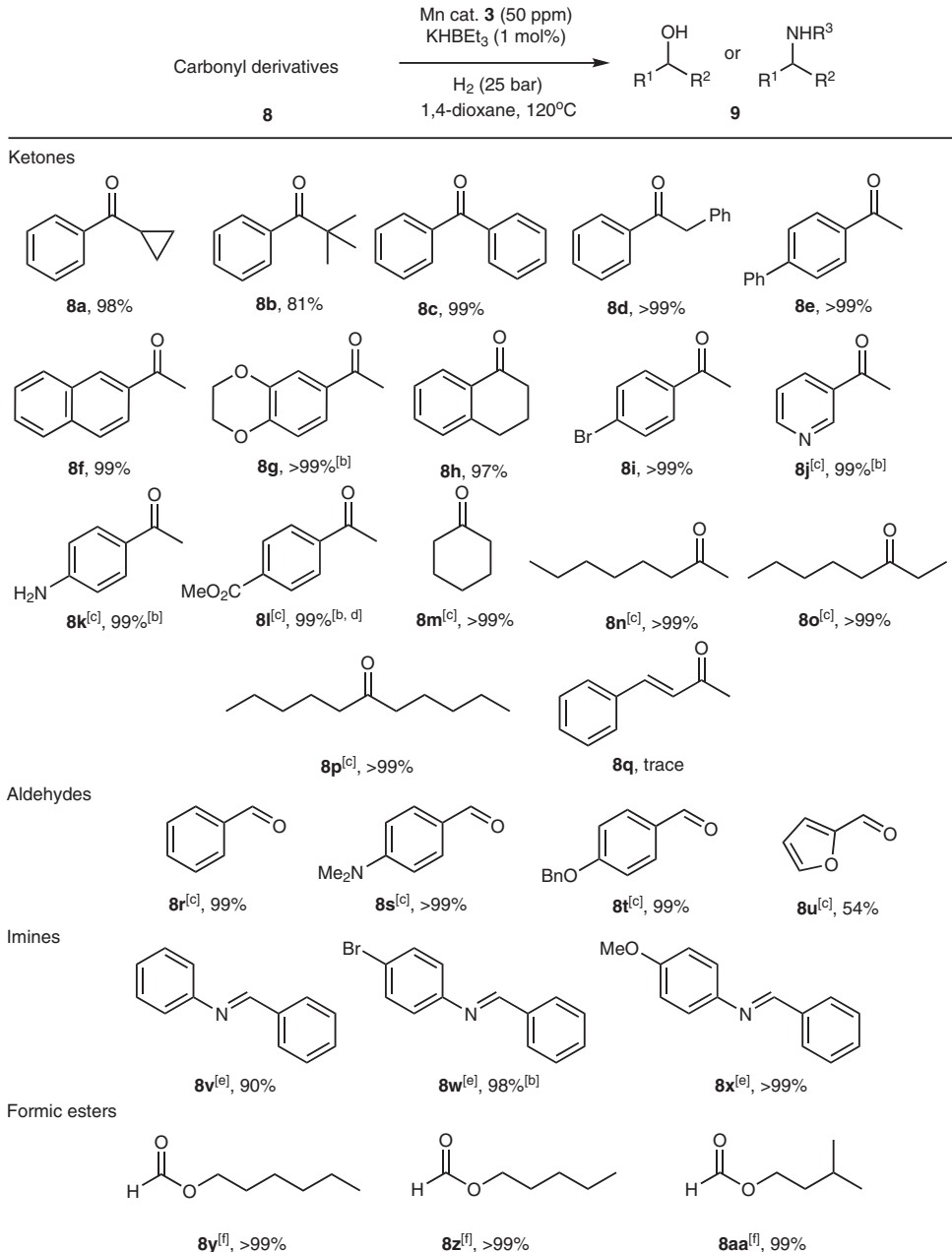

**Fig. 5 Results of catalytic hydrogenation with 3.** [a] Typical conditions: 5 mmol substrate, complex **3** (50 p.p.m.), KBHEt$_3$ (1 mol%) in 1,4-dioxane (3 mL), 120 °C, $P = 25$ bar H$_2$, 24 h. Yields determined by GC-FID with dodecane internal standard. [b] Isolated yields. [c] Reactions carried out in isopropanol (3 mL) at 80 °C instead. [d] The product was corresponding isopropyl ester identified by NMR. [e] 100 p.p.m. of **3** used in isopropanol (3 mL), 120 °C. [f] 200 p.p.m. of **3** used.

A noteworthy exception to this set was conjugated α,β-unsaturated ketone **8q** that was not converted by our catalytic system at appreciable level. In addition to ketones, functionalized aldehydes (**8r–8u**) and imines (**8v–8x**) were converted using 50–100 p.p.m. catalyst loading. Finally, the activity of **3** was sufficient to convert formate esters (**8y–8aa**) to the corresponding alcohols at 200 p.p.m. catalyst loading.

In summary, our findings highlight a Mn(I)–CNP complex **3** as a truly robust and versatile hydrogenation catalyst. A large part of its performance stems from the unusual coordination behavior of the tridentate ligand in complex **3** that opens up the catalyst activation pathways that are unavailable for conventional Mn pincers. As an outcome, one obtains a highly stable catalyst

tolerating high reaction temperatures, while operating at p.p.m.-level loadings. Apart from ketones, the catalytic system allows for the efficient reduction of various unsaturated functional groups, including aldehydes, imines, and formic acid esters in quantitative yields. With the mechanistic analysis of the catalytic action of **3** underway, its performance highlights the high potential of manganese for hydrogenation catalysis. With the introduction of robust mixed donor ligand systems, we anticipate developments in this dynamic field.

## Methods

**General procedure for catalytic hydrogenation**. Stock solutions of **3** (0.01 M) were prepared in dioxane solvent. In a typical run, substrate (5 mmol), dioxane

(3 mL), dodecane internal standard (56.8 μL, 0.25 mmol), base promoter (0.05 mmol), and complex **3** were combined in a 4 mL glass vials and transferred into a stainless steel autoclave in the glovebox. The system was purged with $N_2$ ($3 \times 8$ bar) and $H_2$ ($1 \times 30$ bar), pressurized with $H_2$ to specified pressure, and heated to specified temperature. The yields of products were determined by GC or GC–MS.

### Kinetic study of acetophenone hydrogenation with Mn catalyst 3 on a large scale.

Inside the glovebox, a stock solution of **3** (0.0125 M) was prepared in 0.875 mL dioxane, treated with 0.125 mL of 1 M $KBHEt_3$ solution in THF and stirred for 0.5 h. A 1 mL syringe was loaded with complex **3** (500, 250, 100, and 50 μL) and $KBHEt_3$ (62.5, 93.8, 112.5, and 118.8 μL) in dioxane (total volume 0.7 mL), and a 20 mL syringe was loaded with acetophenone (1.460 mL, 12.5 mmol) and dodecane (113.6 μL, 0.625 mmol) in 10 mL dioxane. Under $N_2$ flow, the substrate syringe was first injected into high pressure stainless steel reactor, in which a glass liner was inserted in advance. The dissolved catalyst was then placed in an injection port and the system was purged with $H_2$ ($3 \times 10$ bar). The reactor was brought to at 120 °C at 50 bar $H_2$ pressure with stirring at 500 r.p.m. and reaction was initiated by injecting the catalyst solution. The samples were withdrawn at given time intervals using an autosampler apparatus and analyzed with GC. Data plotted in Fig. 4 of the manuscript.

### Data availability

Data relating to the synthetic procedures, materials and characterization, optimization studies, DFT calculations, $H_2$ consumption traces, and spectral data are available in the Supplementary Information. CCDC-1994375 contains the supplementary crystallographic data for this paper. These data can be obtained free of charge from The Cambridge Crystallographic Data Centre via www.ccdc.cam.ac.uk/data_request/cif. All data generated and analyzed during this study are included in this article and its Supporting Information, and also available from the authors upon reasonable request. Source data are provided with this paper.

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

## Acknowledgements

This research was supported by the European Research Council under the European Union's Horizon 2020 research and innovation program (grant agreement no. 725686). G.A.F. acknowledges NWO for an individual Veni grant. All authors acknowledge SURFsara for computational resources and BT Mass Spectrometry Facility at TU Delft for HRMS measurement.

## Author contributions

W.Y., G.A.F., and E.A.P. wrote the manuscript; W.Y. designed, conducted the experiments, and analyzed the data; I.Y.C. performed the DFT calculation; R.K.A.S. conducted part of kinetics studies; M.W. and C.M. performed crystal structure determination and analysis; G.A.F. and E.A.P. supervised the research.

## Competing interests

The authors declare no competing interests.
