## [Peer Review File · Nature Communications]

Reviewers' comments:

Reviewer #1 (Remarks to the Author):

This paper by Filonenko, Pidko and collaborators describes the synthesis and characterization of a Mn(I) complex bearing a tridentate CNP-ligand and its use for catalytic hydrogenation of ketones, imines and formate esters. Additionally, some reactivity studies were performed for the Mn-compound and the kinetics of the reaction was also explored, but presumed hemilability was not actually demonstrated to occur during catalysis nor a mechanistic proposal, relevant in this case, was invoked. Thus, the findings reported in the title manuscript do not represent earthshaking novelty, but interesting incremental work on Mn(I)-catalyzed the reduction of polar double bonds with the added value of using precatalyst in very low amounts, though under harsh conditions, namely, rather high hydrogen pressure. I do not recommend this paper for publication in Nature, but suitable for journals like Organometallics, Dalton Trans or ChemCatChem provided the authors address the following remarks:

1. In the introduction on page 2, lines 38 and 39: reported data are misleading, nitriles were not hydrogenated under 10 bar H₂ with 1 mol% [Mn] at 60 °C, but under 50 bar H₂ with 3 mol% [Mn] at 120 °C. Data for ketones is also reported incorrectly.
2. Table 1, entry 4 and Page 6, line 101: specify the structural or condensed formula for A. Why is the point of making a comparison between 3 and A in this part of the paper?
3. Page 6, lines 104 to 106: do the authors know how catalyst is actually deactivated when used at "very low catalytic loadings"? Is this a decomposition pathway or the formation of an inert species in solution?
4. Authors are highly encouraged to perform control reactions such as poisoning with Hg and use of monodentate phosphines to support a homogeneous reaction is actually taking place.
5. Control reaction with corresponding N-methylated analog of precatalyst 3 is missing and must be performed to support or discard an outer-sphere mechanism featuring metal-ligand cooperation.
6. Figure S12: In caption, change "13C-NMR" to "13C{1H}-NMR". Also, signals for CO and Mn-C(NHC) are not really distinguishable, neither assignable since they seem to lay beyond detection limit. Consider and correct this.
7. Page 7, lines 123 to 127: crystal structures (in the solid state) are not representative of phenomena occurring in solution. The fact that angles are the ones reported in the cited lines does not necessarily mean or imply tridentate ligand is specifically hemilabile.
8. Page 7, line 136: how the authors calculated a 24% conversion value during formation of 5? Specify.
9. Page 8, line 138: Please be clear on what two isomers are proposed. Figure S35 only shows one structure calculated by DFT, which is the same displayed in the main text, so it is not clear enough how the other structure looks like.
10. Page 9, line 160; Figure S35: difference between 6 and 6b is that in 6b phosphine moiety is rotated through a sigma-bond. Is this actually originating a difference, $\Delta\delta\text{H} = 0.48$, between two distinguishable 1H-NMR hydride signals? If NMR acquisition was performed at rt, it does not seem logical to observe two resolved signals. In such case, what is the rotation barrier for that sigma-bond and how VT-NMR spectra look like? What about isomers with hydride ligands located trans to a CO or to the carbon atom of the NHC? Also consider this remark for answering point 9.

11. Page 9, line 166: hemilability was indeed observed by NMR, however, its role was not elucidated during catalytic turnover. Phosphine de-coordinated so a thermodynamically favored Mn-H bond was formed, from there on, no insights about an inner or outer-sphere mechanism were provided, neither the influence of hemilability in both catalytic activity and catalyst presumed stability was assessed. For instance, was 6 recovered or detected at the end of catalysis?

12. Lewis acid BEt₃ is formed in situ as a by-product from reaction between NaHBET₃ and 3, what is its role during catalytic turnover? Can BEt₃ participation be ruled out during catalysis?

13. Page 10, line 183: Where does an induction period value of 15 min come from? Clarify this.

14. In Scheme 3, generic structure 9 is not actually a general structural formula for the products obtained. Modify to make a general one or depict the products obtained instead of the substrates scoped.

Reviewer #2 (Remarks to the Author):

The authors describe a novel Mn hydrogenation catalyst. I feel that the flexible side arm is indeed a beneficial feature and I am impressed by the catalyst stability. Mn-based hydrogenation catalysis had been unknown until 2016 and ever since impressive progress has been made. Unfortunately, our understanding how such Mn catalysts operate and how we can improve their performance rationally is extremely low. The authors try to address this key question in an interesting fashion and, consequently, I am in favor of publishing this beautiful piece of work.

The authors may want to consider the following issues!

I have one key issue: the role of the base for the catalysis beside catalyst activation. During catalysis, significantly more base is needed for top performance than one needs to just activate the catalyst. I miss a detailed study in which the authors investigate the hydrogenation rate depending on the precatalyst being activated with two, three, four, five ... equiv. of the base until no further increase in rate is observed. I am very curious if and at which ratio the performance high is reached. I feel the base is doing more than just activating the catalyst or actually further activating the catalyst. I also want to point the attention of the authors to ref. 6b.

Minor:

With regard to the last sentence, I miss the citation of Fertig et al. ACS Catalysis vol. 8 (2018) . - pp. 8525-8530.

Compound references are untypical with Nature journals

Rhett Kempe

Reviewer #3 (Remarks to the Author):

In this paper, Pidko and co-workers report on the catalytic hydrogenation of carbonyl compounds utilizing a new well-defined Mn(I) catalyst. The very low catalyst loadings are impressive and an improvement in Mn-catalyzed hydrogenation reaction. However, the big advantage of homogeneous catalysis in comparison to heterogeneous catalysis is the higher reactivity under

mild conditions. Performing reactions at $>100^{\circ}\text{C}$ (with very low catalyst loading) furnishes the key point of homogeneous catalysis. Especially if Mn-complexes are already known to reduce ketones at room temperature.

Several aspects arise during reading this manuscript:

- For ligand L1, which is not known to literature, no EA or HRMS is provided in the SI. This also applies for complex 5.
- Ad reaction time Table 1&2 : “[b] Total reaction time and that of GC analysis ...” what does that mean?
- Ad Figure 1 trace D: why are two doublets or a doublet of a doublet present in the spectra? The authors argue that two isomers are formed, but do not give any explanation which isomers are formed. The authors refer to the mixture of isomers as complex 5. Theoretical calculations (Figure S38) propose only one signal, which corresponds to only one isomer. Furthermore, the resonances of Figure 1 trace D do not fit with the ones written in the text (line 137).
- Ad Figure 2 trace A: the authors describe the disappearance of species 6a and 6b upon addition of acetophenone. In the main article, no further information on the formed species is given. In the SI (page 31) a proposed structure of the formed complex 9 is given. In this case, the authors propose a cationic, coordinatively saturated complex with an alkoxide as counterion. How did the authors assign the structure of this complex? An alkoxide as counterion seems very counterintuitive.
- Why is such a high base concentration (usually 1 mol% KOtBu or KEt₃H) necessary if the catalyst loading is <200 ppm? This is a ratio of activator/catalyst of 500 or even higher for lower catalyst loading.
- Scheme 3: Why is scope and limitation done at 25 bar H₂ instead of 50 bar as in Table 2? Why is iPrOH used for several substrates instead of 1,4-dioxane? No blank experiments were done in order to rule out transferhydrogenation. Furthermore, substrate 8k resulted in partially elimination of the generated alcohol, yielding a styrene derivative. The authors stated (in the SI) that this is attributed to the high reaction temperature which furnishes the advantage of low catalyst loading.
- Ad Scheme 3: Most of the catalysis products were not isolated and only GC-yields are given, which is very uncommon especially if $>99\%$ GC-yield is reported. Simple purification via filtration over a short plug of silica would lead to a clean product. Only 4 isolated yields are presented. In addition to that, only NMR codes are given in the SI and no NMR spectra were provided.
- Furthermore, the mass balance is between 90% and 110%. This seems not to be precise enough for giving reliable TONs.
- Line 225: the authors state, that esters can be hydrogenated. However formic esters could be hydrogenated, but the ester functionality in substrate 8l (not a formic ester) was not reduced. Only transesterification was observed.

Additional comments on the manuscript itself:

1. The language of the manuscript (and especially of the SI) is not suitable for such a high-impact journal.

- Wrong referencing of Garcia and Kirchner (line 47)! Reference 10 refers to the work of Kirchner and coworkers and Reference 11 to Garcia and coworker. Reference 11 appears again in line 153. Is the work of Kirchner or Garcia referenced?
- Milstein and coworkers published several articles (e.g. *Angew. Chem. Int. Ed.* 2017, 56, 4229-4233) in which the central donor in a PNP-based Mn(I) is hemilabile. Another work was performed with a PNN-based Mn(I) system. These complexes were used of dehydrogenation reactions and at least one of those articles should be cited in this context, since they are operating via a closely related concept.
- Ad Scheme 2: the benzene rings in L1 and 3 look very distorted.

In my opinion the chemistry is described here is very interesting but not of sufficient interest to the readers of *Nature Commun.* and I recommend this contribution for publication in a more specialized journal.

We are happy to provide point-by-point reply to the referees comments. Our replies are marked in blue and changes introduced to revised manuscript are also highlighted.

Reviewer # 1:

This paper by Filonenko, Pidko and collaborators describes the synthesis and characterization of a Mn(I) complex bearing a tridentate CNP-ligand and its use for catalytic hydrogenation of ketones, imines and formate esters. Additionally, some reactivity studies were performed for the Mn-compound and the kinetics of the reaction was also explored, but presumed hemilability was not actually demonstrated to occur during catalysis nor a mechanistic proposal, relevant in this case, was invoked. Thus, the findings reported in the title manuscript do not represent earthshaking novelty, but interesting incremental work on Mn(I)-catalyzed the reduction of polar double bonds with the added value of using precatalyst in very low amounts, though under harsh conditions, namely, rather high hydrogen pressure. I do not recommend this paper for publication in Nature, but suitable for journals like Organometallics, Dalton Trans or ChemCatChem provided the authors address the following remarks:

Response: We thank the referee for taking the time to review our work. We disagree with the reasoning regarding the harshness of reaction conditions as a measure of potency of any catalytic system. Temperature and pressure are reaction parameters that are meaningless if not viewed together with catalyst loading and reaction time. We intentionally aimed at operating at higher temperatures to demonstrate the stability of the catalytic system under these practically relevant conditions when the majority of Mn catalysts undergo deactivation. While temperatures around 100°C are most industrially relevant, our work also features temperatures of 60 – 120°C with some of the experimentation done at 10-50 bar H₂ pressure, thus, on the "mild" end of the conditions window.

To clarify this point, we extended the introduction section of the revised manuscript and accounted for the points raised by the reviewer.

1. In the introduction on page 2, lines 38 and 39: reported data are misleading, nitriles were not hydrogenated under 10 bar H₂ with 1 mol% [Mn] at 60 °C, but under 50 bar H₂ with 3 mol% [Mn] at 120 °C. Data for ketones is also reported incorrectly.

Response: We thank the referee for noticing this. The description on page 2 has been changed to: "at 1-3 mol% loading at 60-120 °C and 10-50 bar H₂ pressure" in the revised manuscript.

2. Table 1, entry 4 and Page 6, line 101: specify the structural or condensed formula for A. Why is the point of making a comparison between 3 and A in this part of the paper?

Response: We thank the reviewer for the comment. The condensed formula has been added in revised manuscript. Catalyst A, which could operate at 120 °C, is among the

most thermally stable Mn catalysts reported to date. Since we performed reactions at high temperature, one might argue that any Mn catalyst would have excellent performance at high temperature. To clarify this we compared 3 and A to show that deactivation of common catalysts (of which A is an excellent example) indeed inhibits catalysis.

3. Page 6, lines 104 to 106: do the authors know how catalyst is actually deactivated when used at "very low catalytic loadings"? Is this a decomposition pathway or the formation of an inert species in solution?

Response: We propose that similar to most other homogeneous catalysts, decomposition is the main contributor to catalyst deactivation at high temperature. Deactivation at sub-200 ppm catalyst loadings can be observed for a variety of Mn catalysts if one performs kinetic investigation. The data describing this phenomenon is scarce with the sole report on Mn catalysis disclosed by our group. We, therefore, cannot make strong assumptions regarding deactivation mechanism currently.

4. Authors are highly encouraged to perform control reactions such as poisoning with Hg and use of monodentate phosphines to support a homogeneous reaction is actually taking place.

Response: We appreciate the reviewer's comment. A control reaction with excess Hg has been carried out and we found no poisoning by Hg suggesting that catalysis is indeed homogeneous. The new data is placed in Table 2, Page 10 of the revised manuscript.

5. Control reaction with corresponding N-methylated analog of precatalyst 3 is missing and must be performed to support or discard an outer-sphere mechanism featuring metal-ligand cooperation.

Response: The reviewer might be confusing the two concepts. Metal-ligand cooperation and outer sphere mechanisms are two separate entities and can take place together or independently (see *ACIE*, 2015, 54, 12236). The methylated complex test alone doesn't answer the mechanistic question posed by the reviewer. An entirely new investigation would be necessary to elaborate on that.

6. Figure S12: In caption, change "¹³C-NMR" to "¹³C{¹H}-NMR". Also, signals for CO and Mn-C(NHC) are not really distinguishable, neither assignable since they seem to lay beyond detection limit. Consider and correct this.

Response: We appreciate the reviewer's comments. The captions of Figures S3, S6 and S12 have been corrected. The presence of CO ligands in complex 4 is confirmed independently by IR spectroscopy and X-ray crystallography. It is not uncommon for carbonyl and NHC carbon resonances to be broadened in ¹³C NMR, especially for labile complexes in coordinating solvents. We collected the new NMR data and a new spectrum is provided in the revised manuscript.

7. Page 7, lines 123 to 127: crystal structures (in the solid state) are not representative of phenomena occurring in solution. The fact that angles are the ones reported in the cited lines does not necessarily mean or imply tridentate ligand is specifically hemilabile.

Response: The reviewer is correct. We did not make this claim but merely pointed to *potential* hemilability of this catalyst implied by a large strain evidenced by diffraction data. The confirmation of hemilability was done further in the manuscript during the discussion of the activation data. The reviewer acknowledges this fact in his/her point No. 11 of this letter.

8. Page 7, line 136: how the authors calculated a 24% conversion value during formation of 5? Specify.

Response: This estimation was done on the basis of NMR integration. Complex 5 was the main detectible product in reaction of complex 4 with H₂. Integral intensity of newly formed hydride peaks (-3.46, -3.49 ppm) was compared to that of the methylene resonance of 4 (4.80 ppm) to obtain the conversion estimate. We clarify this in more detail in section S3 of the Supporting Information.

9. Page 8, line 138: Please be clear on what two isomers are proposed. Figure S35 only shows one structure calculated by DFT, which is the same displayed in the main text, so it is not clear enough how the other structure looks like.

Response: Conformational analysis has been carried out in the DFT study and the most representative structures were included in the original discussion. We are happy to full set of analyzed isomers in the SI (Revised section S8) of revised manuscript together with additional computational details and discussion. Additional clarification is included in the manuscript, all figure captions and Figures 1 and 2 are corrected.

10. Page 9, line 160; Figure S35: difference between 6 and 6b is that in 6b phosphine moiety is rotated through a sigma-bond. Is this actually originating a difference, $\Delta\delta\text{H} = 0.48$, between two distinguishable ¹H-NMR hydride signals? If NMR acquisition was performed at rt, it does not seem logical to observe two resolved signals. In such case, what is the rotation barrier for that sigma-bond and how VT-NMR spectra look like? What about isomers with hydride ligands located trans to a CO or to the carbon atom of the NHC? Also consider this remark for answering point 9.

Response: The assignment was done on the basis of DFT calculations. We agree with the referee. We observed no active exchange between the hydride resonances at elevated temperature that rules out 6b structure as an alternative. We performed new calculations to screen the possible isomers of 5 and 6 and rule out the positional hydride isomers where hydride is located trans to NHC unit, phosphine donor and N-donor ligands. We could confirm that meridionally bound ligand should be present in both 5 and 6 and hydride ligands are likely placed trans to CO as suggested by the referee. The analyzed structures are provided in reply to point 9 of the referee and

additional discussion on the isomer formation is given in the revised manuscript and section S8 of the SI.

11. Page 9, line 166: hemilability was indeed observed by NMR, however, its role was not elucidated during catalytic turnover. Phosphine de-coordinated so a thermodynamically favored Mn-H bond was formed, from there on, no insights about an inner or outer-sphere mechanism were provided, neither the influence of hemilability in both catalytic activity and catalyst presumed stability was assessed. For instance, was 6 recovered or detected at the end of catalysis?

Response: The points mentioned by the referee are actually in agreement with our statement that hemilability plays a prominent role in catalyst activation. In the absence of hemilability we observed manifold lower catalytic activity as shown in Figure 2 of the manuscript. We stress that these data highlight the importance of hemilabile activation of Mn precatalyst and any attempt at assigning the specific mechanism to the catalytic reaction would be speculative and preliminary at this point. As we use catalyst species at 5-100 ppm loading, recovery of those complexes after catalysis is extremely challenging. The mechanistic question regarding inner vs outer sphere operation is still open, as it is for the majority of the hydrogenation catalysts, and deserves an entirely new investigation.

12. Lewis acid BEt_3 is formed in situ as a by-product from reaction between NaHBEt_3 and 3, what is its role during catalytic turnover? Can BEt_3 participation be ruled out during catalysis?

Response: We thank the referee for this suggestion. We examined our data and found no evidence for Lewis acid interaction with the catalyst. We assumed that triethylboron may interact with Lewis basic groups within MnCNP complex. We examined our data and found no difference in ^1H and ^{31}P data for complex 4, having negatively charged N-donor and potentially hemilabile P-donor, in the absence and in the presence of BEt_3 (See Figure S10 and S19 of the SI). On the level of stoichiometric reactions, the Lewis acid appears dormant, however, elucidating its potential role is a focus of an ongoing mechanistic investigation.

13. Page 10, line 183: Where does an induction period value of 15 min come from? Clarify this.

Response: This point is discussed on page 7 of the manuscript. Namely, we relate the induction period to the Mn-H formation that is a slow process in the absence of hemilability. In line with this suggestion, our data on the same figure suggests that the catalyst activation involving hemilability effectively eliminates the induction period (See insert in Figure 2)

14. In Scheme 3, generic structure 9 is not actually a general structural formula for the products obtained. Modify to make a general one or depict the products obtained instead of the substrates scoped.

Response: We appreciate the reviewer's comments. The generic structure of 9 has been changed to:

Reviewer # 2 (Remarks to the Author):

The authors describe a novel Mn hydrogenation catalyst. I feel that the flexible side arm is indeed a beneficial feature and I am impressed by the catalyst stability. Mn-based hydrogenation catalysis had been unknown until 2016 and ever since impressive progress has been made. Unfortunately, our understanding how such Mn catalysts operate and how we can improve their performance rationally is extremely low. The authors try to address this key question in an interesting fashion and, consequently, I am in favor of publishing this beautiful piece of work.

The authors may want to consider the following issues!

I have one key issue: the role of the base for the catalysis beside catalyst activation. During catalysis, significantly more base is needed for top performance than one needs to just activate the catalyst. I miss a detailed study in which the authors investigate the hydrogenation rate depending on the precatalyst being activated with two, three, four, five equiv. of the base until no further increase in rate is observed. I am very curious if and at which ratio the performance high is reached. I feel the base is doing more than just activating the catalyst or actually further activating the catalyst. I also want to point the attention of the authors to ref. 6b.

Response: We thank the referee for this suggestion. The requested tests were conducted and we can confirm that there is indeed a threshold in base or hydride donor concentration necessary to promote catalysis (See Figure S33). At above 500 equiv. of promoter we observe no further improvement of catalytic activity and below that value catalytic activity drops notably. Due to the fact that any substrate and solvent used in catalysis by default contains ppm levels of water we believe that trace impurities might be one of the reasons for a super stoichiometric amount of promoter needed. For example, an aliquot of dioxane used in catalytic test at best contains 3 ppm of water that amounts to 5 μmol of water in the reactor in the best case scenario from the solvent alone. The promoter loading threshold at which the observed activity deteriorates was found to be 0.1 mol %, a value corresponding to 12.5 μmol of promoter per run. Considering the similarity, we can assume that elimination of water and other reactive impurities might be one of the functions of the base or hydride donor promoters. These features are critically important for operation at low catalyst loadings and we appreciate the interest of the Reviewer. Additional tests with KO^tBu amount variation has been carried out, which also indicate similar threshold (see Figure S32).

Minor:

Wirth regard of the last sentence, I miss the citation of Fertig et al. ACS Catalysis vol. 8 (2018) . - pp. 8525-8530.

Response: We thank the reviewer for this suggestion. This citation has been added to the revised manuscript.

Compound references are untypical with Nature journals

Rhett Kempe

Reviewer # 3 (Remarks to the Author):

In this paper, Pidko and co-workers report on the catalytic hydrogenation of carbonyl compounds utilizing a new well-defined Mn(I) catalyst. The very low catalyst loadings are impressive and an improvement in Mn-catalyzed hydrogenation reaction. However, the big advantage of homogeneous catalysis in comparison to heterogeneous catalysis is the higher reactivity under mild conditions. Performing reactions at $>100^{\circ}\text{C}$ (with very low catalyst loading) furnishes the key point of homogeneous catalysis. Especially if Mn-complexes are already known to reduce ketones at room temperature.

Response: We thank the referee for this point. We strongly disagree with this interpretation of the benefits of homogeneous catalysis. Low temperature of operation is a consequence of high intrinsic activity, but the temperature alone is a meaningless parameter. When we consider economics, low catalyst loadings and high reaction temperatures are favoured for the sake of time and material costs. At high temperatures reactions proceed faster and the temperature of water steam ($90\text{-}150^{\circ}\text{C}$) is the most industrially relevant one, however, Mn catalysts rarely survive high reaction temperatures at low catalyst loadings. We countered this behaviour using the new catalyst design and activation approach that brings the activity on par with noble metal catalysts while being unique from the chemistry standpoint.

1. For ligand L1, which is not known to literature, no EA or HRMS is provided in the SI. This also applies for complex 5.

Response: We thank the reviewer for this suggestion. The ligand L1 is additionally characterized with HRMS. Due to its reactive nature, complex 5 cannot be characterized with common MS techniques as it would not ionize in the ESI source and rapidly decompose in MALDI setting. We have obtained additional EA data for a monohydrate of 5. The additional characterization data have been provided in the SI of revised manuscript.

2. Ad reaction time Table 1&2 : "[b] Total reaction time and that of GC analysis ..." what does that mean?

Response: This is the time at which the autoclaves were depressurized and final yields of all reactions were determined. We presented an H_2 pressure tracking during

all catalytic runs and specified the time of GC analysis performed at the end of the reaction.

3. Ad Figure 1 trace D: why are two doublets or a doublet of a doublet present in the spectra? The authors argue that two isomers are formed, but do not give any explanation which isomers are formed. The authors refer to the mixture of isomers as complex 5. Theoretical calculations (Figure S38) propose only one signal, which corresponds to only one isomer. Furthermore, the resonances of Figure 1 trace D do not fit with the ones written in the text (line 137).

Response: We thank the Referee for this suggestion. Additional description of suggested structures of isomers of 5 and 6 is included in the revised manuscript together with additional DFT data requested previously by Referee 1 that is placed in section S8 of the SI.

4. Ad Figure 2 trace A: the authors describe the disappearance of species 6a and 6b upon addition of acetophenone. In the main article, no further information on the formed species is given. In the SI (page 31) a proposed structure of the formed complex 9 is given. In this case, the authors propose a cationic, coordinatively saturated complex with an alkoxide as counterion. How did the authors assign the structure of this complex? An alkoxide as counterion seems very counterintuitive.

Response: Our main reason for performing this reaction was to demonstrate that complex 6 can indeed transfer hydride to acetophenone. The structure of 9 was merely suggested based on the observation of the carbonyl bands in IR spectra at positions typical for cationic Mn complexes as well as ^{31}P NMR data suggesting the same. We restrict this description to the SI as the *proposed* species 9 are formed in a complex reaction mixture that prevents its isolation. We added a more detailed description of reaction with acetophenone to accurately depict our observations in the SI in section S3.

5. Why is such a high base concentration (usually 1 mol% KOtBu or KEt₃H) necessary if the catalyst loading is <200 ppm? This is a ratio of activator/catalyst of 500 or even higher for lower catalyst loading.

Response: As noted by Referee 2, the question of the base involvement is still debated across hydrogenation literature. KHBET₃ is a highly reactive compound that immediately reduced ketones and is not present in its original form throughout the catalysis. The necessary minimum loadings for KHBET₃ were found to be 0.05 mol%. Importantly, KO^tBu, that does not reduce ketones reaches rate saturation at this concentration. New data are added to Figure S32 and S33 of the revised manuscript.

6. Scheme 3: Why is scope and limitation done at 25 bar H₂ instead of 50 bar as in Table 2?

Response: These conditions are practically milder. We tested the influence of H₂ pressure (10-50 bar, see Figure S33) and found that once $p(\text{H}_2) > 20$ bar the system

showed catalytic activity sufficient for full conversion. For this reason we decided to apply lower H₂ pressure for substrate scope.

7. Why is iPrOH used for several substrates instead of 1,4-dioxane? No blank experiments were done in order to rule out transferhydrogenation.

Response: The substrates reduced in iPrOH are activated aromatic ketones, alkyl ketones, and aldehydes, that could not be fully hydrogenated at standard conditions (1,4-dioxane, 24 h). Our optimization indicated iPrOH is a better solvent that can provide full conversion and reduction selectivity. As suggested by the referee we performed the necessary blank reactions and found that transfer hydrogenation does not make major contribution to carbonyl reduction although trace activity in the absence of H₂ can indeed be observed. The new data is included in Table S4 of the revised manuscript.

8. Furthermore, substrate 8k resulted in partially elimination of the generated alcohol, yielding a styrene derivative. The authors stated (in the SI) that this is attributed to the high reaction temperature which furnishes the advantage of low catalyst loading.

Response: We appreciate the reviewer's comments. The temperature mentioned is *not the reaction temperature* but that of GC-MS inlet flange. The elimination in 1-(4-aminophenyl)ethanol at high inlet temperature (300 °C) was observed previously by Panarin (Russ. J. Gen. Chem. 2010, 80, 1309-1313). To avoid confusion we repeated the experiment and obtained and isolated yield of 99% after silica pad filtration. The corrected results and spectral data for 8k are given in revised manuscript.

9. Ad Scheme 3: Most of the catalysis products were not isolated and only GC-yields are given, which is very uncommon especially if >99% GC-yield is reported. Simple purification via filtration over a short plug of silica would lead to a clean product. Only 4 isolated yields are presented. In addition to that, only NMR codes are given in the SI and no NMR spectra were provided.

Response: We appreciate the reviewer's comments. The substrate scope is used to assess the activity of a new Mn catalyst. We opted not to isolate commercially available alcohols and amines that have no specific sensitivity to common workup procedures. However, for products 8g, 8j, 8k, 8l, 8w, which could not be identified by GC-MS, we did the isolation workup as the most reliable technique. The full NMR spectra are included in the supporting information of the revised version of the manuscript.

10. Furthermore, the mass balance is between 90% and 110%. This seems not to be precise enough for giving reliable TONs.

Response: We appreciate the reviewer's comments. The standard deviation in internal standard readings was recalculated to be within 2.7% interval. The uncertainty is now included in the TON estimates.

11. Line 225: the authors state, that esters can be hydrogenated. However formic esters could be hydrogenation, but the ester functionality in substrate 8I (not a formic ester) was not reduced. Only transesterification was observed.

Response: We appreciate the reviewer's comment. All the ester substrates are referred to as "formic acid esters" in revised manuscript.

Additional comments on the manuscript itself:

1. The language of the manuscript (and especially of the SI) is not suitable for such a high-impact journal.

Response: We cannot comment on that point and leave it for consideration of the editorial team.

2. Wrong referencing of Garcia and Kirchner (line 47)! Reference 10 refers to the work of Kirchner and coworkers and Reference 11 to Garcia and coworker. Reference 11 appears again in line 153. Is the work of Kirchner or Garcia referenced?

Response: We have corrected referencing in revised manuscript. The reference 11 in line 153 is Garcia's work.

3. Milstein and coworkers published several articles (e.g. Angew. Chem. Int. Ed. 2017, 56, 4229-4233) in which the central donor in a PNP-based Mn(I) is hemilabile. Another work was performed with a PNN-based Mn(I) system. These complexes were used of dehydrogenation reactions and at least one of those articles should be cited in this context, since they are operating via a closely related concept.

Response: We thank the referee for this suggestion. We have added Milstein and coworkers' work (Angew. Chem. Int. Ed. 2017, 56, 4229-4233, Angew. Chem. Int. Ed. 2017, 56, 14992 –14996) to the references 17 of revised manuscript.

4. Ad Scheme 2: the benzene rings in L1 and 3 look very distorted.

Response: We appreciate the reviewer's comments. The representations have been improved in the revised manuscript.

In my opinion the chemistry is described here is very interesting but not of sufficient interest to the readers of Nature Commun. and I recommend this contribution for publication in a more specialized journal.

Response: We appreciate the time spent reviewing our work and kindly disagree with the judgement of the referee.

REVIEWER COMMENTS

Reviewer #1 (Remarks to the Author):

After reviewing the changes made to this manuscript by Filonenko, Pidko and collaborators, I found the following:

1. Hemilability is not clearly demonstrated as a key factor for catalysis. Such concept is included in the title so it is expected from the authors to put some effort to prove how hemilability is responsible for outstanding catalytic activity. Otherwise, there is enough room for speculation as in many other publications.
2. I disagree with the authors when they say that temperature and pressure are meaningless, they are critical variables in catalytic performance: is waste of energy meaningless? These variables should always receive attention.
3. Hg drop test was not performed under identical reaction conditions than the optimized reaction (i.e. Table 2, entry 2 vs Table 2, entry 7), so no conclusive evidence on the homogenous nature of the reaction can be given so far.
4. Authors said that captions of Fig. S3, S6, and S12 were corrected, but they were not, probably as a consequence of a careless revision. Additionally, there are other wrong captions in the SI, which I will not specify this time since authors did not take care of details in the preparation of this revised version of the manuscript.

Confirming what I stated in the latter revision of this work, I do not recommend this for publication in Nature but in other journals such as Organometallics or ChemCatChem.

Reviewer #4 (Remarks to the Author):

Dear Editor,

The submitted manuscript describes the synthesis of a new tridentate PNC ligand for Mn-catalysed hydrogenation reactions. The obtained catalyst is particularly robust and active allowing very high turnover at low catalyst loadings in various applications. For manganese catalysed hydrogenation reactions this represents an improvement over the state of the art catalytic systems.

In particular, the authors studied the formation of a catalytic competent hydride species from precursor **3** upon activation either with base/hydrogen or with a (basic) boron-hydride species and have found that the latter method results in faster generation of a hydride manganese complex and, thus, faster hydrogenation with almost no induction period.

Saying that, the authors assert from the title, over the abstract and then throughout the manuscript that hemilability is responsible for the high catalyst performances and suggest hemilability as a ligand design strategy for future research.

The point is that no prove for hemilability behavior of the ligand during catalysis are provided and most probably this does not occur at all.

By treatment of **4** with hydrogen not the supposed "labile" PPh²-arm but a CO ligand is displaced generating **5**. The ligand coordination in both (DFT) suggested isomers of **5** is not facial but meridional: thus, the ring strain observed in **4**, which has been thought to be responsible for a "potential lability" (page 7 line 7), will be released upon conversion of 4 to 5. All the speculation on the hemilability relies on **6**, which is formed from **3** upon treatment with KHBet₃, which has indeed an open phosphine arm.

This species, however, converts to **5** upon irreversible displacement of CO. This process is slow at room temperature, but under used catalytic conditions i.e. at 80 to 120°C will certainly be much more rapid.

And then? Does the hydrogenation mechanism require to detach the phosphine arm again or rather, like in the outer-sphere mechanism, this process is not necessary at all? To invoke hemilability as design element imparting excellent stability and activity is not demonstrated and superfluous: a tridentate pincer ligand is able to stabilize the Mn-center too and even minor changes in the ligand structure can account for increased catalytic performances.

I warmly suggest the authors not to stress hemilability as the explanation for the catalytic performances. The results will be not diminished thereby. As stated by the authors in the point-by-point reply to referee 1, the hemilability plays a role, if any, during the activation not during the catalysis. The extended stability of catalysts at low loadings maybe due to the very large excess of KBHET₃ used, whereas an excess of KO^tBu is detrimental cf. Figure 2 (B). This may explain the difference in activity and stability between the two activation methods at low catalyst loading, where the excess of the activation reagents is particularly pronounced.

I do not suggest the publication of the manuscript in the present form.

Remarks:

1. Title: eliminate hemilability

2. Abstract: add formate esters; eliminate hemilability

3. Page 1, line 32: hydrogenation catalysts based on osmium play a very minor role. Much more important are rhodium-based catalysts.

4. Page 1, line 34: reference 3. Add this recently published book chapter: Clarke, Matthew L.; Widegren, Magnus B.; Hydrogenation Reactions Using Group III to Group VII Transition Metals" in Homogeneous Hydrogenation with Non-Precious Catalysts Edited by Teichert, Johannes F (2020), 111-140.

5. The main species upon treatment of **3** with KBHET₃ is not **6** but **4**, as can be clearly seen in the ³¹P NMR spectrum (S23): the authors should correct Figure 2 (A) and mention this in the manuscript. Thus, KBHET₃ does not act only as hydrogenation reagent but also, and mainly, as a base. Just showing in Figure 2 (A) the hydride region of the ¹H NMR is misleading.

6. Complex **3** and **4**: Upon coordination, no free rotation of the mesityl group is possible anymore (also in solution) as shown by the two distinct peaks for the ortho methyl substituents in ¹H and ¹³C NMR. This observation is probably worth to be mentioned in the manuscript.

Supporting information:

7. Ligand **1**: ¹H-NMR. The purity of this compound seems below 95% and, consequently, the integrals are rather approximate (e.g. the ortho methyl groups of the mesityl substituent). The PF₆ signal is missing in ³¹P-NMR.

We are happy to provide point-by-point reply to the referees comments. Our replies are marked in blue and changes introduced to revised manuscript are also highlighted.

Reviewer # 1:

1. Hemilability is not clearly demonstrated as a key factor for catalysis. Such concept is included in the title so it is expected from the authors to put some effort to prove how hemilability is responsible for outstanding catalytic activity. Otherwise, there is enough room for speculation as in many other publications.

Response: We appreciate the comment of the referee. Following the suggestion of the Editor and Referee 4 we remove the phrasing that implies the involvement of hemilability in the *catalytic cycle*. We thank the referee for referring to the catalytic activity as "outstanding". In the revised manuscript, as also proposed by Referee 4, we limit the discussion of hemilability as a feature of the activation procedure, which helps to eliminate the induction period and provides competent hydride species in part due to the hemilabile ligand behaviour. As noted by the Referee 4, the catalytic cycle does not necessarily proceed via the hemilabile pathway, and we confirm that with additional mechanistic experiments included in the revised manuscript. Namely, we confirm that catalytic turnover leads to the gradual generation of *both* hydride species 5 and 6 suggesting that hemilability is involved in the catalytic turnover in a limited way. The new data is presented in the revised Figure 2A. The new text on pages 10-11 reads:

"As our data depicted in Figure 2A suggested that both 5 and 6 are exhibiting the hydride transfer reactivity upon the contact with acetophenone, we further attempted to observe the outcome of the catalytic turnover on the relative composition of the reaction mixture using NMR spectroscopy. Results of this experiment are presented in Figure 2A. We observed that hydrogenation of acetophenone substrate with reaction mixtures containing predominantly hydride species 6 results in the accumulation of the dicarbonyl complex 5. While suggesting that the catalytic turnover involving solely species 6 and associated ligand hemilability is possible, gradual accumulation of 5 in the course of several catalytic turnovers suggests that the hydrogenation can likely proceed over the complex 5 at low catalyst loadings."

3. Hg drop test was not performed under identical reaction conditions than the optimized reaction (i.e. Table 2, entry 2 vs Table 2, entry 7), so no conclusive evidence on the homogenous nature of the reaction can be given so far.

Response: We understand the concern of the referee that 12 h reaction time was used in the control experiments. The control reaction has been repeated with identical reaction time (3h) and we found no poisoning by Hg suggesting that catalysis is indeed homogeneous. The new data is placed in Table 2, Page 11 of the revised manuscript.

4. Authors said that captions of Fig. S3, S6, and S12 were corrected, but they were not, probably as a consequence of a careless revision. Additionally, there are other

wrong captions in the SI, which I will not specify this time since authors did not take care of details in the preparation of this revised version of the manuscript.

Response: We appreciate the comment of the referee and apologize for misinterpretation. We actually did change the spectral data in our previous submission – all axis captions for all carbon NMR spectra were changed as requested to $^1\text{H}\{^{13}\text{C}\}$. In current revised SI we also modify figure captions as requested. Additional comment that ^{13}C and ^{31}P NMR data are proton decoupled is included in the general method description.

Reviewer # 4:

The submitted manuscript describes the synthesis of a new tridentate PNC ligand for Mn-catalysed hydrogenation reactions. The obtained catalyst is particularly robust and active allowing very high turnover at low catalyst loadings in various applications. For manganese catalysed hydrogenation reactions this represents an improvement over the state of the art catalytic systems. In particular, the authors studied the formation of a catalytic competent hydride species from precursor 3 upon activation either with base/hydrogen or with a (basic) boron-hydride species and have found that the latter method results in faster generation of a hydride manganese complex and, thus, faster hydrogenation with almost no induction period.

We thank the Referee for the positive feedback.

Saying that, the authors assert from the title, over the abstract and then throughout the manuscript that hemilability is responsible for the high catalyst performances and suggest hemilability as a ligand design strategy for future research. The point is that no prove for hemilability behavior of the ligand during catalysis are provided and most probably this does not occur at all. By treatment of 4 with hydrogen not the supposed "labile" PPh²-arm but a CO ligand is displaced generating 5. The ligand coordination in both (DFT) suggested isomers of 5 is not facial but meridional: thus, the ring strain observed in 4, which has been thought to be responsible for a "potential lability" (page 7 line 7), will be released upon conversion of 4 to 5.

Response: We thank the referee for the comments and agree that high strain in 4 would not necessarily translate to derived meridional complexes. The hemilability reference on page 8 is removed.

All the speculation on the hemilability relies on 6, which is formed from 3 upon treatment with KHBET₃, which has indeed an open phosphine arm. This species, however, converts to 5 upon irreversible displacement of CO. This process is slow at room temperature, but under used catalytic conditions i.e. at 80 to 120°C will certainly be much more rapid.

And then? Does the hydrogenation mechanism require to detach the phosphine arm again or rather, like in the outer-sphere mechanism, this process is not necessary at all? To invoke hemilability as design element imparting excellent stability and activity is not demonstrated and superfluous: a tridentate pincer ligand is able to stabilize the Mn-center too and even minor changes in the ligand structure can account for increased catalytic performances.

Response: We appreciate the assertion of the referee. We do agree that complex 6 could covert to 5 under the catalytic conditions and prevent phosphine hemilability from taking place during the catalytic cycle. Additional experiments described in response to Referee 1 (comment 1) confirm this suggestion. We indeed observe accumulation of complex 5 as catalytic reaction proceeds from the reaction mixture containing complex 6 predominantly. We thank the Referee for this valuable suggestion and incorporate these findings in the revised manuscript.

I warmly suggest the authors not to stress hemilability as the explanation for the catalytic performances. The results will be not diminished thereby. As stated by the authors in the point-by-point reply to referee 1, the hemilability plays a role, if any, during the activation not during the catalysis. The extended stability of catalysts at low loadings maybe due to the very large excess of KBHET_3 used, whereas an excess of KO^tBu is detrimental cf. Figure 2 (B). This may explain the difference in activity and stability between the two activation methods at low catalyst loading, where the excess of the activation reagents is particularly pronounced.

Response: We thank the referee for this suggestion. We recognize how the stress on hemilability can be unnecessary and correct the revised manuscript accordingly.

I do not suggest the publication of the manuscript in the present form.

We thank the Referee for careful evaluation and hope that the revised manuscript would be suitable for publication.

Remarks:

1. Title: eliminate hemilability
2. Abstract: add formate esters; eliminate hemilability

Response: Abstract and title of the revised manuscript were corrected as requested.

3. Page 1, line 32: hydrogenation catalysts based on osmium play a very minor role. Much more important are rhodium-based catalysts.

Response: We thank the referee for the suggestion. Relevant literature on rhodium-catalysed hydrogenations is added to the revised manuscript (new References 2b-c)

4. Page 1, line 34: reference 3. Add this recently published book chapter: Clarke, Matthew L.; Widgren, Magnus B.; Hydrogenation Reactions Using Group III to Group VII Transition Metals" in Homogeneous Hydrogenation with Non-Precious Catalysts Edited by Teichert, Johannes F (2020), 111-140.

Response: We thank the referee for this suggestion. We have added this chapter to Reference 3 of the revised manuscript.

5. The main species upon treatment of 3 with KBHET_3 is not 6 but 4, as can be clearly seen in the ^{31}P NMR spectrum (S23): the authors should correct Figure 2 (A) and mention this in the manuscript. Thus, KBHET_3 does not act only as hydrogenation reagent but also, and mainly, as a base. Just showing in Figure 2 (A) the hydride region of the ^1H NMR is misleading.

Response: We thank the referee for the comments. The Figure 2A has been revised and main product 4 has been mentioned in revised manuscript on page 8, line 153. All selected hydride data is labelled to indicate relative content of the species and full spectra are given in the SI. The new text on page 8 reads:

"In THF- d_8 at room temperature the reaction of 3 with KBHET₃ readily yields a reaction mixture containing 69 % of the amido complex 4 with the remainder comprised of new manganese hydride species 6 (Figure 2A) that exist as a mixture of isomers."

6. Complex 3 and 4: Upon coordination, no free rotation of the mesityl group is possible anymore (also in solution) as shown by the two distinct peaks for the ortho methyl substituents in ¹H and ¹³C NMR. This observation is probably worth to be mentioned in the manuscript.

Response: We thank the referee for this suggestion. Additional description is added to page 7, line 122 of the revised manuscript.

Supporting information:

7. Ligand 1: ¹H-NMR. The purity of this compound seems below 95% and, consequently, the integrals are rather approximate (e.g. the ortho methyl groups of the mesityl substituent). The PF₆ signal is missing in ³¹P-NMR.

Response: We thank the referee for this comment. The integration values were off due to the incorrect baseline treatment. We reprocessed the spectrum that is now shown with cumulative integration curves. Same treatment was applied to all ¹H data in the revised SI. New ³¹P-NMR spectrum containing PF₆ signal has also been added in the revised Supporting Information (Figure S2). We added a statement that L1 was used without additional purification for synthesis of 3 that is analytically pure.

REVIEWERS' COMMENTS

Reviewer #4 (Remarks to the Author):

The revised manuscript can be published as it is. Just "Osmium" (page 2, line 33) and ref 2c should be deleted.

We are happy to provide point-by-point reply to the referees comments. Our replies are marked in blue.

Reviewer # 4:

The revised manuscript can be published as it is. Just "Osmium" (page 2, line 33) and ref 2c should be deleted.

The requested changes are done in the revised manuscript.